# χSPN: Characteristic Interventional Sum-Product Networks for Causal Inference in Hybrid Domains

**Harsh Poonia**[1]     **Moritz Willig**[2]     **Zhongjie Yu**[2]     **Matej Zečević**[2]     **Kristian Kersting**[2,3,4]     **Devendra Singh Dhami**[5]

[1]Indian Institute of Technology Bombay, India
[2]Technical University of Darmstadt, Darmstadt, Germany,
[3]Hessian Center for Artificial Intelligence (hessian.AI), Darmstadt, Germany,
[4]German Research Center for Artificial Intelligence (DFKI), Darmstadt, Germany,
[5]Eindhoven University of Technology, Eindhoven, Netherlands,

## Abstract

Causal inference in hybrid domains, characterized by a mixture of discrete and continuous variables, presents a formidable challenge. We take a step towards this direction and propose **Ch**aracteristic **I**nterventional Sum-Product Network (χSPN) that is capable of estimating interventional distributions in presence of random variables drawn from mixed distributions. χSPN uses characteristic functions in the leaves of an interventional SPN (iSPN) thereby providing a unified view for discrete and continuous random variables through the Fourier–Stieltjes transform of the probability measures. A neural network is used to estimate the parameters of the learned iSPN using the intervened data. Our experiments on 3 synthetic heterogeneous datasets suggest that χSPN can effectively capture the interventional distributions for both discrete and continuous variables while being expressive and causally adequate. We also show that χSPN generalize to multiple interventions while being trained only on single intervention data.

## 1 INTRODUCTION

Most real-world data, irrespective of the underlying domain, consists of variables originating from multiple distributions such as continuous, discrete and/or categorical. In the realm of statistical modeling, understanding and accurately characterizing such data poses formidable challenges. Mixed distributions, arising from the amalgamation of distinct sub-populations within a dataset, exhibit a complexity that traditional statistical methodologies often struggle to capture. This can lead to machine learning models becoming either inapplicable or producing incorrect results during inference. This applies not only to correlation-based methods but can also have adverse effects on causality-based methods.

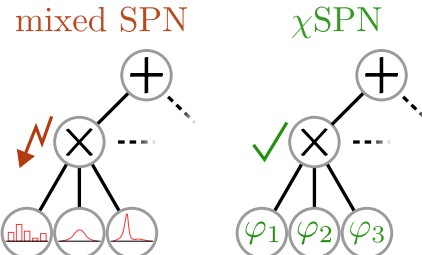

Figure 1: **Correct Mixing of Distributions via χSPN.** Classical mixedSPN naïvely multiply discrete probabilities and continuous densities, leading to an ill-defined probability measure. For practical applications, large density values can possibly outweigh normalized discrete probabilities, biasing parameter estimation. χSPN overcome this problem by transforming discrete and continuous variables into a shared spectral domain. Sum and product operations on the spectral representations are well defined. (Best viewed in color.)

Causal inference [Pearl, 2009, Spirtes, 2010], the study of cause-and-effect relationships, is a fundamental pursuit in statistics, yet its application to mixed distributions introduces unique challenges. Whether the data stems from diverse demographic groups, heterogeneous environments, or multifaceted systems, the ability to infer causal relationships in presence of mixed distributions is crucial for advancing our understanding of causal phenomena in various real-world domains.

Several probabilistic methods, such as hybrid Bayesian networks [Monti and Cooper, 1998, Murphy, 1998], Gaussian-Ising mixed model [Lauritzen et al., 1989, Cheng et al., 2017] and variants of Markov random fields [Fahrmeir and Lang, 2001, Fridman, 2003], have been proposed to handle hybrid domains. A major drawback of these methods is the difficulty of inference [Lerner and Parr, 2001] as it can quickly become intractable. This becomes a glaring issue in widespread adoption of these methods for causal inference which in itself is a challenging problem [Peters et al., 2017]. To overcome the problem of intractable inference, a lot of

work has been done on probabilistic circuits (PCs) [Jaeger, 2004, Lowd and Domingos, 2012], specifically sum-product networks [Domingos and Poon, 2012, Gens and Domingos, 2013], which guarantee inference in linear time under some specific conditions. There have been efforts within the PCs to handle hybrid data with development of methods such as mixed SPN [Molina et al., 2018] and Bayesian SPN [Trapp et al., 2019]. These methods are restrictive as they fail to model leaves with distributions that do not have closed-form density expressions and rely on a histogram or density function representation of the probability measures. A recent approach [Yu et al., 2023], proposes the use of characteristic functions to provide a unified formalization of distributions over heterogeneous data in the spectral domain.

SPNs have also been successfully applied to different rungs of the causal ladder [Zečević et al., 2021, Busch et al., 2023]. We specifically consider interventional SPN (iSPN) [Zečević et al., 2021] that learns interventional distributions using SPNs over-parameterized by neural networks. In this work, we propose $\chi$SPN (**Ch**aracteristic **I**nterventional Sum-Product Networks), the first causal models that are capable of efficiently inferring causal quantities i.e., interventional distributions in presence of mixed data. Methods such as mixedSPN naïvely multiply discrete probabilities and continuous densities, leading to an ill-defined probability measure as large density values can possibly outweigh normalized discrete probabilities biasing parameter estimation. $\chi$SPN overcome this problem by transforming discrete and continuous variables into a shared spectral domain using characteristic functions in the leaves of iSPN (see Fig.1). Overall, we make the following important contributions:

1. We present the first causal model capable of performing inference on hybrid domains in a tractable fashion.

2. We demonstrate the effectiveness of combining characteristic functions with iSPN's to naturally handle mixed data i.e. data containing random variables with discrete and continuous distributions.

3. We show that $\chi$SPN can generalize to multiple interventions without any retraining.

We make our code publicly available at: https://github.com/harpoonix/chi-SPN. We will proceed as follows: we first present the required preliminaries and discuss the related work and then define $\chi$SPN. We then present extensive experiments on mixed domains before concluding.

# 2 PRELIMINARIES & RELATED WORK

Before diving into the proposed $\chi$SPN model, we present some necessary background on SPNs, causal models and characteristic functions.

## 2.1 SUM-PRODUCT NETWORKS

Sum-Product Networks [Domingos and Poon, 2012] are a class of deep tractable models, which belong to the family of probabilistic circuits. SPNs facilitate a wide range of exact and efficient inference routines. In particular, marginalisation and conditioning can be done in time which is linear in the size of the network [Zhao et al., 2015, Peharz et al., 2015]. Formally, an SPN is a rooted directed acyclic graph, comprising of sum, product and leaf (or distribution) nodes to encode joint probability distributions $p(\mathbf{X})$. Given an SPN $\mathcal{S} = (G, \mathbf{w})$ with positive parameters $\mathbf{w}$ and a DAG $G = (V, E)$, the values at sum (S) and product (P) nodes can be computed by

$$S(\mathbf{x}) = \sum_{C \in \text{ch}(S)} \mathbf{w}_C C(\mathbf{x}) \quad P(\mathbf{x}) = \prod_{C \in \text{ch}(P)} C(\mathbf{x}) \quad (1)$$

where $\text{ch}(P)$ are the children of $P$. The SPN outputs are computed at the root node, $S_R(\mathbf{x})$. The scope of a leaf node is the random variable $X$ that it models. The scope of an internal node is the union of scopes of all its children. SPNs satisfy the properties of *completeness* and *decomposability*. An SPN $\mathcal{S}$ is complete if for every sum node $u$ in $\mathcal{S}$ the scopes of its children are all the same. An SPN $\mathcal{S}$ is decomposable if for every product node $u$ in $\mathcal{S}$ the scopes of its children are pairwise disjoint.

Gated or Conditional SPNs are deep tractable models for estimating multivariate conditional densities $p(Y|x)$ [Shao et al., 2022], by conditioning the parameters of vanilla SPNs on the input using DNNs as gate functions. They introduce gating nodes where the weights $g_i(X)$ are parameterized by the provided evidence $X$ to encode functional dependencies on the input.

## 2.2 CAUSAL MODELS

Structural Causal Models provide a framework to formalize a notion of causality via graphical models [Pearl, 2009].

**Definition 2.1** (SCM). *A structural causal model is a tuple* $\mathcal{M} := \langle \mathbf{V}, \mathbf{U}, \mathbf{F}, P_{\mathbf{U}} \rangle$ *over a set of variables* $\mathbf{X} = \{X_1, \ldots, X_K\}$ *taking values in* $\boldsymbol{\mathcal{X}} = \prod_{k \in \{1 \ldots K\}} \mathcal{X}_k$ *subject to a strict partial order* $<_{\mathbf{x}}$, *where*

- $\mathbf{V} = \{X_1, \ldots, X_N\} \subseteq \mathbf{X}, N \leq K$ *is the set of endogenous variables,*

- $\mathbf{U} = \mathbf{X} \setminus \mathbf{V} = \{X_{N+1}, \ldots, X_K\}$ *is the set of exogenous variables,*

- $\mathbf{F} = \{f_1, \ldots, f_N\}$ *is the set of deterministic structural equations, i. e.* $V_i := f_i(\mathbf{X}')$ *for* $V_i \in \mathbf{V}$ *and* $\mathbf{X}' \subseteq \{X_j \in \mathbf{X} | X_j <_{\mathbf{x}} V_i\}$,

- $P_{\mathbf{U}}$ *is the probability distribution over the exogenous variables* $\mathbf{U}$.

The relationships between the variables as described by $\mathbf{F}$ induce the directed graph $G(\mathcal{M})$ which by definition is acyclic due to $<_{\mathbf{X}}$. The exogenous variables $\mathbf{U}$ are usually unobserved. We say that an SCM $\mathcal{M}$ entails the probability distribution $P_{\mathbf{V}}^{\mathcal{M}}$ over the set of endogenous variables $\mathbf{V}$.

Interventions $do(X)$ change the way variables are determined by replacing their respective structural equation $f_i$. In particular perfect interventions $do(X_i = v)$ replace the unintervened $f_i$ by the constant assignment $X_i := v$. Every intervention induces a new intervened graph $G(\mathcal{M}_{do(V_i=v_i)})$ to which we will refer to as $\hat{G}$ for notational brevity. Likewise, every intervened causal model $\mathcal{M}_{do(V_i=v_i)}$ entails a new probability distribution $P_{\mathbf{V}}^{\mathcal{M}_{do(V_i=v_i)}}$.

Often times only a subset of all possible interventions is considered. If not silently omitted, this restriction can be made explicit by modeling SCM with a set of *allowed interventions* $\mathcal{I}$ [Halpern, 2000, Beckers and Halpern, 2019, Rubenstein et al., 2017]. In this paper, we will usually evaluate our models over the set of single perfect interventions:

$$\mathcal{I} = \{\{do(X_i = v_i)\}\}_{i \subseteq \{1...N\}}. \tag{2}$$

Note that for further practical application of our model, training is not restricted to any particular choice of $\mathcal{I}$. We provide additional evaluations inspecting multi-intervention generalization of the model.

**Probabilistic Circuits and Causality.** Several types of probabilistic models exist as of today that allow for varying degrees of tractable inference. Classical SCM as extensions of Bayesian Networks [Pearl, 1985] as well as their neural realizations [Xia et al., 2021] suffer from #P-hard time complexity for exact (and NP-hard complexity for approximate) inference [Eiter and Lukasiewicz, 2002]. To alleviate parts of this problem, other model choices such as normalizing flows [Papamakarios et al., 2021] are picked to approximate the causal distributions [Khemakhem et al., 2021, Melnychuk et al., 2023, Javaloy et al., 2023]. These models, however, are not able to perform tractable marginal inference. When required to perform such queries, Sum-Product-Networks pose a suitable model class.

### 2.3 CHARACTERISTIC FUNCTIONS

Characteristic functions (CF) provide a unified view for discrete and continuous RVs through the Fourier–Stieltjes transform of their probability measures. Let $\boldsymbol{X} \in \mathbb{R}^d$ be a random vector, the CF of $\boldsymbol{X}$ for $\boldsymbol{t} \in \mathbb{R}^d$ is given as:

$$\varphi_{\boldsymbol{X}}(\boldsymbol{t}) = \mathbb{E}\left[\exp\left(\mathrm{i}\boldsymbol{t}^\top \boldsymbol{X}\right)\right] = \int_{\boldsymbol{x}\in\mathbb{R}^d} \exp\left(\mathrm{i}\boldsymbol{t}^\top \boldsymbol{x}\right) \mu_{\boldsymbol{X}}(\mathrm{d}\boldsymbol{x}), \tag{3}$$

where $\mu_{\boldsymbol{X}}$ is the distribution/probability measure of $\boldsymbol{X}$. The following properties of CFs are relevant for the remaining discussion:

1. $\varphi_X(0) = 1$ and $|\varphi_X(t)| \leq 1$
2. any two RVs $X_1$ and $X_2$ have the same distribution iff $\varphi_{X_1} = \varphi_{X_2}$
3. two RVs $X_1, X_2$ are independent iff $\varphi_{X_1,X_2}(s,t) = \varphi_{X_1}(s)\varphi_{X_2}(t)$

We refer to Sasvári [2013] for more details of CFs.

**Theorem 2.2** (Lévy's inversion theorem [Sasvári, 2013]). *Let $X$ be a real-valued random variable, $\mu_X$ its probability measure, and $\varphi_X : \mathbb{R} \to \mathbb{C}$ its characteristic function. Then for any $a, b \in \mathbb{R}, a < b$, we have that*

$$\lim_{T\to\infty} \frac{1}{2\pi} \int_{-T}^{T} \frac{\exp(-\mathrm{i}ta) - \exp(-\mathrm{i}tb)}{\mathrm{i}t} \varphi_X(t)\mathrm{d}t$$
$$= \mu_X[(a,b)] + \frac{1}{2}\left(\mu_X(a) + \mu_X(b)\right), \tag{4}$$

*and, hence, $\varphi_X$ uniquely determines $\mu_X$.*

**Corollary 2.3.** *If $\int_{\mathbb{R}} |\varphi_X(t)|\,\mathrm{d}t < \infty$, then $X$ has a continuous probability density function $f_x$ given by*

$$f_X(x) = \frac{1}{2\pi} \int_{\mathbb{R}} \exp(-\mathrm{i}tx)\varphi_X(t)\mathrm{d}t. \tag{5}$$

Note that not every probability measure admits an analytical solution to Eq. 5, e.g., only special cases of $\alpha$-stable distributions have a closed-form density function [Nolan, 2013], and numerical integration might be needed.

## 3 $\chi$SPN

We build upon the construction of interventional sum-product networks (iSPN) by Zečević et al. [2021]. We estimate $p\left(V_i \mid do\left(\mathbf{U}_j = \mathbf{u}_j\right)\right)$ by learning a function approximator $f(\mathbf{G}; \boldsymbol{\theta})$ (e.g. neural network), which takes as input the (mutilated) causal graph $\mathbf{G} \in \{0,1\}^{N\times N}$ encoded as an adjacency matrix, to predict the parameters $\psi$ of a SPN $g(\mathbf{D}; \psi)$ that estimates the density of the given data matrix $\{\mathbf{V}_i\}_{i=1}^{K} = \mathbf{D} \in \mathbb{R}^{K\times N}$.

When the iSPN is trained end to end on the log likelihood of the training data, the log densities computed at leaves modeling both discrete and continuous variables are propagated up the network to the root of the SPN. When a common class of leaves is used, say one parameterized by a normal distribution, it acts as a suboptimal way to model discrete variables that do not quite fit this class of normals. Even different distributions at the leaves may not fully be able to capture the joint distribution of a heterogeneous group of variables, since a sum-product combination of different kinds of discrete and continuous densities is likely to result in some variables overshadowing the others in the value computed at the root of the SPN. Moreover, we are restricted to using only those parametric distributions at the leaves that have a closed form density function. This is true only

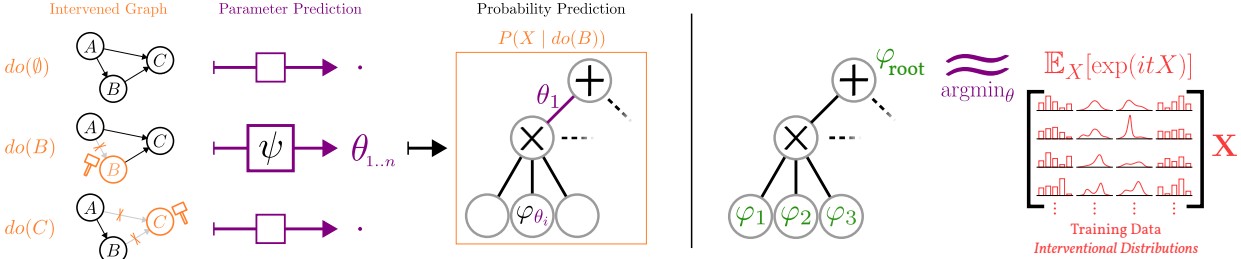

Figure 2: $\chi$**SPN parameters are provided by intervention information (Left).** $\chi$SPN accounts for interventions that change the graph structure and –in consequence– the intervened probability distribution. The parameterization of the SPN leaves and weights ($\theta$) is predicted by a neural network conditioned on intervention information. **Training Setup (Right).** Parameters $\theta$ of the $\chi$SPN are trained by matching the predicted $\chi$ distribution at the root node against the $\chi$ distribution computed from interventional data. (Best viewed in color.)

in the case of special $\alpha$-stable distributions [Nolan, 2013]. We aim to address these problems with our proposed $\chi$SPN that can be defined as follows:

**Definition 3.1** ($\chi$ Sum-Product Network). *A $\chi$SPN $\mathcal{C}$ is the joint model $C(\mathbf{G}, \mathbf{D}) = (g_\varphi, g_\mu)(\mathbf{D}; \boldsymbol{\psi} = f(\mathbf{G}; \boldsymbol{\theta}))$, where $g(\cdot)$ is a SPN that learns the population characteristic function $\varphi$ during training and estimates the interventional density $\mu$ during inference. $f(\cdot)$ is a function approximator and $\boldsymbol{\psi} = f(\mathbf{G})$ are shared parameters.*

A $\chi$SPN is capable of answering interventional queries and, most importantly, allow working with mixed data i.e., where variables of both discrete and continuous distributions are present. Fig. 2(left) shows the overall process of the underlying probability prediction by $\chi$SPN. The parameterization of the $\chi$SPN leaves and weights is predicted by a neural network conditioned on intervention information.

### 3.1 $\chi$SPN STRUCTURE

Inspired by Yu et al. [2023], we modify our iSPN to learn the characteristic function $\varphi_{\boldsymbol{X}}(\boldsymbol{t})$ of the joint density. To this end, we make the leaves of the network learn the CF of a univariate distribution, to model a particular random variable. We modify the calculations at the product and sum nodes as follows.

**Product Nodes.** Decomposability of $\chi$SPN implies that a product node encodes the independence of its children. Let $X$ and $Y$ be two RVs. Following property (3) of CFs, the CF of $X, Y$ is given as $\varphi_{X,Y}(t, s) = \varphi_X(t)\varphi_Y(s)$, if and only if $X$ and $Y$ are independent. Therefore, since the children of a product node all have different scopes, with $\boldsymbol{t} = \bigcup_{N \in ch(P)} \boldsymbol{t}_{sc(N)}$, the characteristic function of product nodes is defined as:

$$\varphi_{\mathrm{P}}(\boldsymbol{t}) = \prod_{\mathrm{N} \in ch(\mathrm{P})} \varphi_{\mathrm{N}}\left(\boldsymbol{t}_{sc(\mathrm{N})}\right), \tag{6}$$

where sc denotes the scope of a node.

**Sum Nodes.** Completeness of $\chi$SPN implies that a sum node encodes the mixture of its children. Let the parameters of $S$ be given as $\sum_{\mathrm{N} \in ch(\mathrm{S})} w_{\mathrm{S,N}} = 1$ and $w_{\mathrm{S,N}} \geq 0, \forall \mathrm{S, N}$. Since all the children of a sum node $S$ have the same scope, the CF at a sum node is:

$$\begin{aligned}\varphi_{\mathrm{S}}(\boldsymbol{t}) &= \int_{\boldsymbol{x} \in \mathbb{R}^d} \exp\left(\mathrm{i}\boldsymbol{t}^\top \boldsymbol{x}\right) \left[\sum_{\mathrm{N} \in ch(\mathrm{S})} w_{\mathrm{S,N}} \mu_{\mathrm{N}}(\mathrm{d}\boldsymbol{x})\right] \\ &= \sum_{\mathrm{N} \in ch(\mathrm{S})} w_{\mathrm{S,N}} \underbrace{\int_{\boldsymbol{x} \in \mathbb{R}^{p_{\mathrm{S}}}} \exp\left(\mathrm{i}\boldsymbol{t}^\top \boldsymbol{x}\right) \mu_{\mathrm{N}}(\mathrm{d}\boldsymbol{x})}_{=\varphi_{\mathrm{N}}(\boldsymbol{t})}.\end{aligned} \tag{7}$$

**Leaf Nodes.** For discrete RVs, we utilize categorical distributions and for continuous RVs, we use $\alpha$-stable distributions. A more detailed discussion on the leaf types can be found in Appendix B.

### 3.2 EXPRESSIVITY

The shared parameters $\psi$ of the $\chi$SPN allow learning of the joint distribution for any dataset $\mathbf{D}_{\hat{G}}$ conditioned on the mutilated causal graph $\hat{G}$, that contains information about the interventions. Neural networks have been shown to act as causal sub-modules e.g. Ke et al. [2019] used a cohort of neural nets to represent a set of structural equations which in turn represent an SCM, providing grounding to the idea of having parameters being estimated from $f$.

The $\chi$SPN also can model any interventional distribution $p_G(\mathbf{V} \mid do(\mathbf{U}))$, permitted by an SCM through interventions to construct the mutilated causal graph $\hat{G}$ by modelling the conditional distribution $p_{\hat{G}}(\mathbf{V} \mid \mathbf{U})$. This follows from Pearl [2009] since $p_G(\mathbf{V}_i = \mathbf{v}_i \mid do(\mathbf{U}_j = \mathbf{u}_j)) = p_{\hat{G}}(\mathbf{V}_i = \mathbf{v}_i \mid \mathbf{U}_j = \mathbf{u}_j)$. The SPN can learn the joint probability $p(X_1 \dots X_n)$ on the $\mathbf{D}_{\hat{G}}$ generated post-intervention and is thus causally adequate. The question of availability of experimental data is an orthogonal one. While in many applications we do not have access to sets of experiments e.g.,

because of monetary or ethical reasons, many other applications in science can in fact provide said sets of experiments e.g., high-throughput biology.

## 3.3 LEARNING

The $\chi$SPN is learned from a set of mixed distribution heterogeneous samples generated from simulating interventions on the underlying SCM. Instead of maximising the log-likelihood, which is not guaranteed to be tractable, we aim to learn the CF for the distribution corresponding to a given intervention. We use the Empirical Characteristic Function (ECF) [Feuerverger and Mureika, 1977] which has been proven to be an unbiased and consistent estimator of the population characteristic function. Given data $\{\boldsymbol{x}_j\}_{j=1}^{n}$ the ECF is given as

$$\hat{\varphi}_{\mathbb{P}}(\boldsymbol{t}) = \frac{1}{n} \sum_{j=1}^{n} \exp\left(\mathrm{i}\boldsymbol{t}^{\top}\boldsymbol{x}_j\right). \quad (8)$$

The overall goal of learning, as shown in Fig. 2(right) is to approximate, as closely as possible, the underlying characteristic function of the intervened data (which we call $\chi$ distribution[1]).

**Evaluation Metric.** A measure of the closeness of two distributions represented by their characteristic functions is the squared characteristic function distance (CFD). The squared CFD between two distributions $\mathbb{P}$ and $\mathbb{Q}$ is defined as

$$\mathrm{CFD}_{\omega}^{2}(\mathbb{P}, \mathbb{Q}) = \int_{\mathbb{R}^d} |\varphi_{\mathbb{P}}(\boldsymbol{t}) - \varphi_{\mathbb{Q}}(\boldsymbol{t})|^2 \, \omega(\boldsymbol{t}; \eta) \mathrm{d}\boldsymbol{t}, \quad (9)$$

where $\omega(\boldsymbol{t}; \eta) > 0$ is a weighting function parameterized by $\eta$ that guarantees the integral in Eq. 9 converges. When $\omega(\boldsymbol{t}; \eta)$ is a probability density function, Eq. 9 can be rewritten as an expectation over $\boldsymbol{t}$ sampled from $\omega$:

$$\mathrm{CFD}_{\omega}^{2}(\mathbb{P}, \mathbb{Q}) = \mathbb{E}_{\boldsymbol{t} \sim \omega(\boldsymbol{t}; \eta)} \left[ |\varphi_{\mathbb{P}}(\boldsymbol{t}) - \varphi_{\mathbb{Q}}(\boldsymbol{t})|^2 \right]. \quad (10)$$

Sriperumbudur et al. [2010] showed that using the uniqueness theorem of CFs, $\mathrm{CFD}_{\omega}(\mathbb{P}, \mathbb{Q}) = 0$ iff $\mathbb{P} = \mathbb{Q}$ which motivates our choice of this distance metric. We refer to Ansari et al. [2020] for a detailed discussion on CFD.

Our learning objective is then to minimise the squared characteristic function distance between the characteristic function estimated at the root of $\chi$SPN and the ECF of the intervened dataset:

---

[1] Not to be confused with the Chi distribution, which is the positive square root of a sum of squared independent Gaussian random variables

$$\mathrm{CFD}^{2}(\mathcal{C}, \hat{\mathbb{P}}_I) = \mathbb{E}_{\boldsymbol{t}}[\varphi_{\mathcal{C}}(\boldsymbol{t}) - \mathbb{E}_{\mathbf{x}_I} \exp\left(\mathrm{i}\boldsymbol{t}^{T}\boldsymbol{x}\right)]^2$$
$$= \frac{1}{k} \sum_{j=1}^{k} \left| \varphi_{\mathcal{C}}\left(\boldsymbol{t}_j\right) - \frac{1}{n} \sum_{i=1}^{n} \exp\left(\mathrm{i}\boldsymbol{t}_j^{\top}\boldsymbol{x}_i\right) \right|^2, \quad (11)$$

where $n$ is the number of data points, $k$ is the number of MCMC samples to estimate the expectation from Eq. 10, and $\boldsymbol{t}_j$ are samples from $\omega(\boldsymbol{t}; \eta)$ which we use $\mathcal{N}\left(\mathbf{0}, \mathrm{diag}\left(\eta^2\right)\right)$ throughout this paper. Applying Sedryakyan's Inequality to Eq. 11, the parameter learning can be operated batch-wise [Yu et al., 2023]. A parameter update step backpropagates through the CFD evaluated on a batch corresponding to single intervention $I$.

It is important to note here that our contribution, in the form of structure and algorithm, for $\chi$SPN isn't a straightforward combination of interventional SPNs with Characteristic Circuits (CCs). The training of CCs and iSPNs are very different. iSPN accepts conditional input about interventions whereas the parameters of a CC are not conditioned on any input. In order to adapt iSPNs to the spectral domain, we need the model to learn the joint interventional density, and for that we make the root of the $\chi$SPN learn the characteristic function of the interventional distribution. We cannot simply introduce characteristic leaves in an iSPN and later transform it into density (through inversion, Section 3.4) for the purpose of learning with log-likelihood of the observed interventional data.

## 3.4 TRACTABILITY OF INFERENCE

Through their recursive nature, $\chi$SPN allow efficient computation of densities in a high dimensional setting even if closed form densities don't exist. To get the joint probability density over all random variables in the SCM, we perform inversion of the characteristic function at the root of the network, for which we use an extension of Corollary 2.3.

**Lemma 3.2** (Inversion). *Let $\mathcal{C}$ be a $\chi$SPN modeling the distribution of RVs $\mathbf{X} = \{X_j\}_{j=1}^{d}$ and employing univariate leaf nodes. If $\int_{\mathbb{R}} |\varphi_{\mathrm{L}}(t)| \, \mathrm{d}t < \infty$ for every leaf L, then $\boldsymbol{X}$ has a continuous probability density function $f_{\boldsymbol{x}}$ given by the integral on the $d$-dimensional space $\mathbb{R}^d$, i.e.,*

$$f_{\boldsymbol{X}}(\boldsymbol{x}) = \frac{1}{(2\pi)^d} \underbrace{\int_{\boldsymbol{t} \in \mathbb{R}^d} \exp\left(-\mathrm{i}\boldsymbol{t}^{\top}\boldsymbol{x}\right) \varphi_{\mathcal{C}}(\boldsymbol{t}) \lambda_d(\mathrm{d}\boldsymbol{t})}_{=\hat{f}_{\mathcal{C}}(\boldsymbol{x})}, \quad (12)$$

*where $\varphi_{\mathcal{C}}(\boldsymbol{t})$ denotes the CF defined by the root of the $\chi$SPN and $\lambda_d$ is the Lebesque measure on $\left(\mathbb{R}^d, \mathcal{B}\left(\mathbb{R}^d\right)\right)$.*

We can recursively compute Eq. 12 for every node. Thus, inversion at every inner node reduces to inversion at its children. We can invoke Corollary 2.3 to obtain density

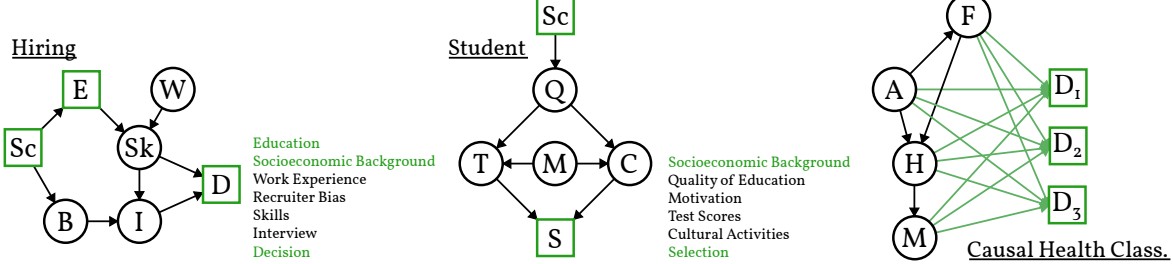

Figure 3: **Evaluated Mixed Datasets.** All $\chi$SPN are trained and evaluated on three mixed type data sets. Hiring and Student data sets contain a mix of continuous (indicated via black circles) and discrete (indicated via green squares) within an exemplary causal process. Causal Health Classification features the important special case of a categorization process resulting in three discrete diagnosis variables which are derived from all-continuous observations. (Best viewed in color.)

measures at leaves, since they model a univariate distribution:

$$\hat{f}_{\mathrm{L}}(x) = 2\pi f_{\mathrm{L}}(x) = \int_{\mathbb{R}} \exp(-\mathrm{i}tx)\varphi_X(t)\lambda(\mathrm{d}t). \quad (13)$$

Let $p_{\mathrm{N}}$ be the number of variables in the scope of node N.

**Sum Nodes.** Using the completeness property of SPNs, for a sum node S:

$$
\begin{aligned}
\hat{f}_{\mathrm{S}}(\boldsymbol{x}) &= \int_{\boldsymbol{t}\in\mathbb{R}^p} \exp\left(-\mathrm{i}\boldsymbol{t}^\top\boldsymbol{x}\right) \varphi_{\mathrm{S}}(\boldsymbol{t})\lambda_p(\mathrm{d}\boldsymbol{t}) \\
&= \sum_{\mathrm{N}\in\mathrm{ch}(\mathrm{S})} w_{\mathrm{S},\mathrm{N}} \underbrace{\int_{\boldsymbol{t}\in\mathbb{R}^{p_{\mathrm{S}}}} \exp\left(-\mathrm{i}\boldsymbol{t}^\top\boldsymbol{x}\right) \varphi_{\mathrm{N}}(\boldsymbol{t})\lambda_{p_{\mathrm{S}}}(\mathrm{d}\boldsymbol{t})}_{=\hat{f}_{\mathrm{N}}(\boldsymbol{x})},
\end{aligned}
$$
$$(14)$$

which is the weighted sum of inversions at its children.

**Product Nodes.** Using the decomposability property of SPNs, and the fact that a product node P models a product distribution assuming independence among its children,

$$
\begin{aligned}
\hat{f}_{\mathrm{P}}(\boldsymbol{x}) &= \int_{\boldsymbol{t}\in\mathbb{R}^{p_{\mathrm{P}}}} \exp\left(-\mathrm{i}\boldsymbol{t}^\top\boldsymbol{x}\right) \varphi_{\mathrm{P}}(\boldsymbol{t})\lambda_{p_{\mathrm{P}}}(\mathrm{d}\boldsymbol{t}) \\
&= \prod_{\mathrm{N}\in\mathrm{ch}(\mathrm{P})} \underbrace{\int_{s\in\mathbb{R}^{p_{\mathrm{N}}}} \exp\left(-\mathrm{i}s^\top\boldsymbol{x}_{[\mathrm{sc}(\mathrm{N})]}\right) \varphi_{\mathrm{N}}(s)\lambda_{p_{\mathrm{N}}}(\mathrm{d}s)}_{=\hat{f}_{\mathrm{N}}\left(\boldsymbol{x}_{[\mathrm{sc}(\mathrm{N})]}\right)},
\end{aligned}
$$
$$(15)$$

where $\boldsymbol{x}_{[\mathrm{sc}(\mathrm{N})]}$ is the subset of dimensions in $\boldsymbol{x}$ that belong to the scope of its child N. The product appears as an application of Fubini's theorem [Fubini, 1907] for product measures.

Numerical integration may be needed for such measures when there is no closed form density at the leaves. A good approximation technique for the inversion at $\alpha$-stable leaves is the Gauss-Hermite quadrature [Hildebrand, 1987], since the integral over the entire domain $[-\infty, \infty]$ is intractable. We approximate the integral $\int_{\mathbb{R}} \exp(-\mathrm{i}tx)\varphi_X(t)\mathrm{d}t$ with a

weighted sum ($w_i$ of function values at certain sampled points ($t_i$) as

$$\int_{-\infty}^{+\infty} e^{-t^2}\left(\underbrace{e^{t^2}\exp(-\mathrm{i}tx)\varphi_X(t)}_{h(t)}\right)\mathrm{d}t \approx \sum_{i=1}^n w_i h\left(t_i\right),$$
$$(16)$$

where $n$ is the number of sample points used (typically $< 100$). This is tractable since the closed form of $\varphi_X(t)$ and by extension $h(t)$ for the $\alpha$-stable leaf is known.

### 3.5 $\chi$SPN IS A UNIVERSAL FUNCTION APPROXIMATOR

The weights of the $\chi$SPN are parameterised by gating functions and distribution parameters and this allows them to induce universal approximators. By using threshold functions, $\theta^+\mathbb{I}(x_i \geq c) + \theta^- \cdot \neg\mathbb{I}(x_i \geq c), c \in \mathbb{R}$, one can encode testing arithmetic circuits [Choi and Darwiche, 2018] , which are proven universal approximators. This renders $\chi$SPN to be universal approximators by design. Moreover, use of characteristic functions allows the leaves of the network to theoretically model all probability distributions, including those that do not have a density function.

## 4 EXPERIMENTAL EVALUATION

$\chi$SPN are tailored towards handling causal graphs with hybrid data i.e. graphs consisting of random variables drawn from both discrete and continuous distributions. Our experiments thus focus on capturing the interventional distributions within such exemplary causal processes. We aim to answer the following questions:

**(Q1)** Can $\chi$SPN effectively estimate the joint probability of the heterogeneous variables conditioned on arbitrary interventions?

**(Q2)** How well does $\chi$SPN capture individual interventional distributions?

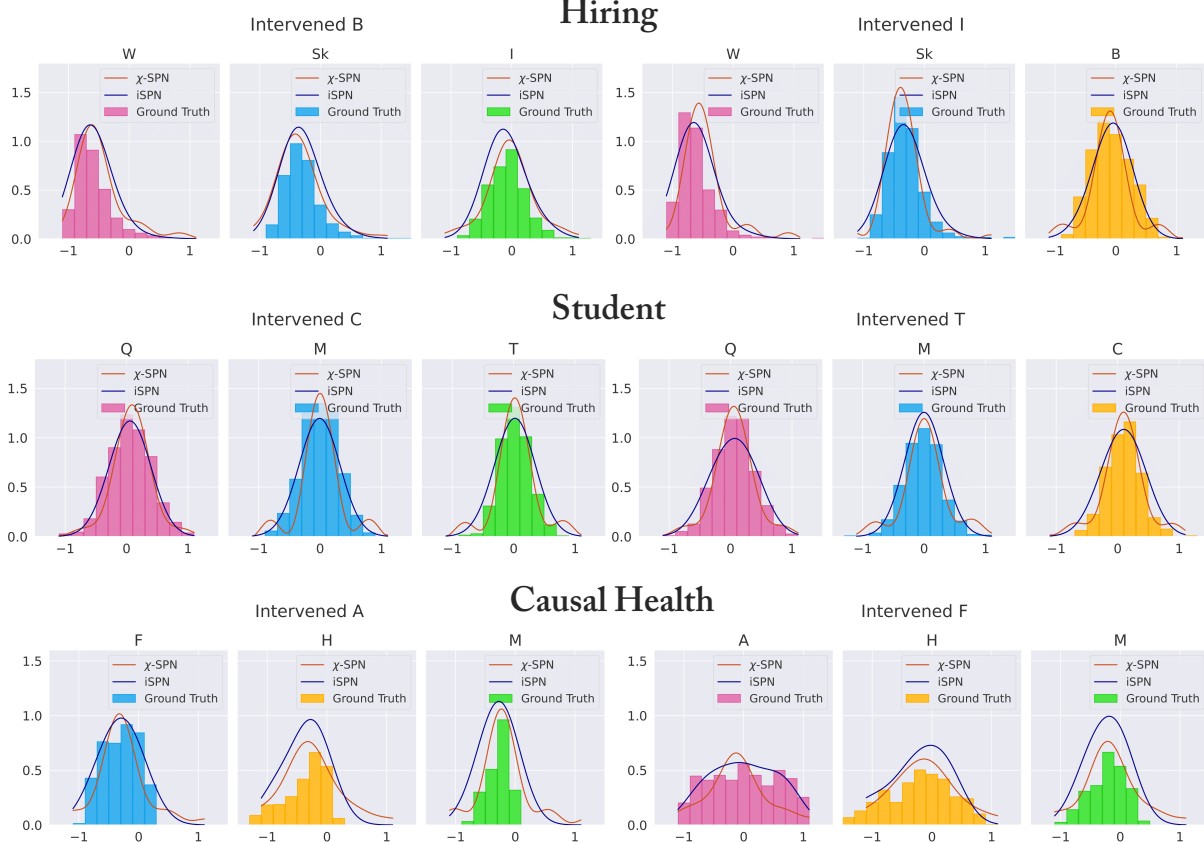

Figure 4: **Approximation of Interventional Densities.** Plots feature the approximated densities of continuous variables for different interventional distributions. Marginalized ground truth distributions (plotted as bar diagrams) and $\chi$SPN approximations (red line) are shown. Modes of the distributions are generally well matched across most plots. Deviations from ground truth show at distribution boundaries as artifacts of the $\chi$ function discretization. (Best viewed in color.)

**(Q3)** Does $\chi$SPN generalize to multiple interventions?

Before presenting our results and answering these questions we briefly describe the data generating process.

**Data Generating Process.** To evaluate $\chi$SPN's ability to model arbitrary interventional distributions on heterogeneous data, we curate 3 synthetic datasets, comprising of an extension to the causal health dataset from Zečević et al. [2021] and two new causal datasets themed around hiring practices and student performance. The SCMs for the 3 datasets are outlined with the variables and their corresponding domains in Fig. 3. We generate 100,000 data points for the extended causal health dataset and 120,000 for the hiring and student datasets. We also use different types of distributions for the noise, such as Gaussian and Pareto noise across all datasets. The train/test split is 80%/20%. A detailed description of the datasets with the underlying distributions can be found in Appendix A.

**Underlying model.** We use the RAT-SPN [Peharz et al., 2020] as a structural base upon which we build our own model. We do not perform structure learning of the $\chi$SPN

and instead adopt a randomised splitting strategy at the nodes. We chose RAT-SPN as it follows a simple randomized procedure for structuring an SPN, thus providing a significant computational advantage over explicit structure learning. The shared parameters $\psi$ of the $\chi$SPN are predicted from a fully connected neural network with 2 hidden layers, with different final layer for gate and leaf parameters. We choose $n = 50$ as the number of sample points used in the Gauss-Hermite quadrature.

**Capability of $\chi$SPN for handling hybrid domains (Q1).** We test our $\chi$SPN on the 3 synthetic datasets described above with interventions on both discrete and continuous variables. We compare $\chi$SPN with the baseline model of iSPN [Zečević et al., 2021] for the continuous case. Fig. 4 shows the captured interventional distributions for the continuous variables. Note that, due to lack of space, we show only 2 variables per dataset here with the complete results shown in the Appendix D in Figs. 9, 10 and 11. It can be seen that our $\chi$SPN captures the modes of the distributions well matched across the datasets. Deviations from ground truth at the distribution boundaries can be attributed to the

| **Causal Health Classification** | | | | | **Hiring** | | | | **Student** | |
| Interv. | D1 | D2 | D3 | | Interv. | E | D | | Interv. | S |
|---|---|---|---|---|---|---|---|---|---|---|
| Observ. | 76.8% | 69.4% | 53.8% | | Observ. | 64.2% | 85.9% | | Observ. | 58.6% |
| do(A) | 76.6% | 69.6% | 53.8% | | do(I) | 64.1% | 65.2% | | do(Q) | 56.5% |
| do(F) | 83.3% | 63.0% | 53.7% | | do(E) | N/A | 84.0% | | do(M) | 54.6% |
| do(H) | 78.9% | 64.0% | 57.0% | | do(Sk) | 64.5% | 61.1% | | do(C) | 59.5% |
| do(M) | 84.7% | 46.0% | 69.3% | | do(B) | 64.6% | 86.7% | | do(T) | 56.4% |

Figure 5: **Accuracies of Discrete Variable Prediction.** Tables contain the prediction accuracies over all discrete variables of the data sets. Results for observational and interventions on the remaining (unintervened) continuous variables are presented.

artifacts of the $\chi$ function discretization. The baseline iSPN generally over or underestimates the underlying distributions. This is expected since iSPN's cannot handle mixed distributions.

Furthermore, Fig. 5 shows the accuracies of discrete variable prediction after intervention on the continuous variables. Since we consider the discrete variables to be the target variables we calculate the accuracies of the discrete value being correctly predicted. Please note that the discrete variables are not always binary. For instance, in the hiring dataset the variable E (education) can take 7 values and in the student dataset the variable S (selection) can take 3 values. For education (E) prediction in the hiring dataset, the top-3 accuracy is reported. The results show that $\chi$SPN can effectively capture the discrete variable values after intervening on then continuous variables. We can thus answer (Q1) affirmatively: $\chi$SPN can handle interventions on mixed datasets, thereby making them applicable to hybrid domains.

**Observational vs Interventional (Q2).** In the following we inspect the ability of our $\chi$SPN to truthfully capture individual interventional distributions. Depending on the number of variables in a graph, every individual interventional distribution is only seen in a small fraction of the samples. For the Hiring and Causal Health Classification, which both contain 7 variables, each intervention is only visible in $\sim 14\%$ of the training data. Even within these samples all causal mechanisms –except the intervened one– are computed in the standard 'observational' behaviour, increasing bias towards observational behaviour. While being powerful density approximators, there exists a chance that $\chi$SPN overfit to the observational distribution in practice for the stated reasons.

Comparing observational (Fig. 4) and interventional (Fig. 8 in the appendix) density estimates we find that no strong degradation in performance is observed when switching from observational to interventional estimation. In particular, the modes of all distributions seem to be estimated matched well. For Hiring and Causal Health data sets slight biases in mode prediction (Hiring variables W, Sk, I; Causal Health variables F, H) are learned for observational data. Qualita-

tively, we find that the discretization artifacts –present at boundaries of the interventional distributions– are strongly reduced for observational data. Like previously, this can be a consequence of the higher supply of observational data.

For discrete variables we compare observational results in the first rows of Fig. 5 against the accuracies of intervention graphs. Recall, that intervening on any variable in the graph does change the actual underlying distribution. Depending on the intervened variable, predictions might, therefore, become easier or harder to predict. While accuracies vary across different interventions, we observe no severe performance degradations. Overall, we can now answer (Q2): $\chi$SPN do not suffer from sampling bias towards observation data and predict interventional distributions equally well.

**Multiple Interventions (Q3).** To test the generalization capabilities of $\chi$SPN, we test the model trained on a single intervention to estimate the distributions captured from multiple interventions. Fig. 6 shows the results for the Student dataset with the application of intervention on two variables simultaneously. For example, the top left 2 graphs show capture the distributions after intervention on the cultural activities and test scores variables (C,T) while the top right 2 graphs capture the distributions after intervention on motivation and cultural activities (M,C). As it can be seen $\chi$SPN is able to faithfully capture the interventional distributions thereby showing the generalization capability of our approach. Additional results, found in Fig. 7, for the Hiring dataset confirm this finding. As before, there are slight biases in the captured distributions but can be attributed to the discretization of the characteristic function. This answers (Q3) affirmatively: $\chi$SPN can efficiently generalize to multiple interventions.

A question might arise here: how does $\chi$SPN generalize to unseen interventions? As the SCM structure is not enforced within the NN, relations have to be learned from a suitable data presentation. Clearly, if a variable is never intervened, the NN can not correlate the input of the intervention signal to an appropriate weight vector for the SPN. Assuming that interventions on a variable are trained, but novel values are observed, general assumptions about neural networks extrapolating/generalizing to novel out-of-distribution values

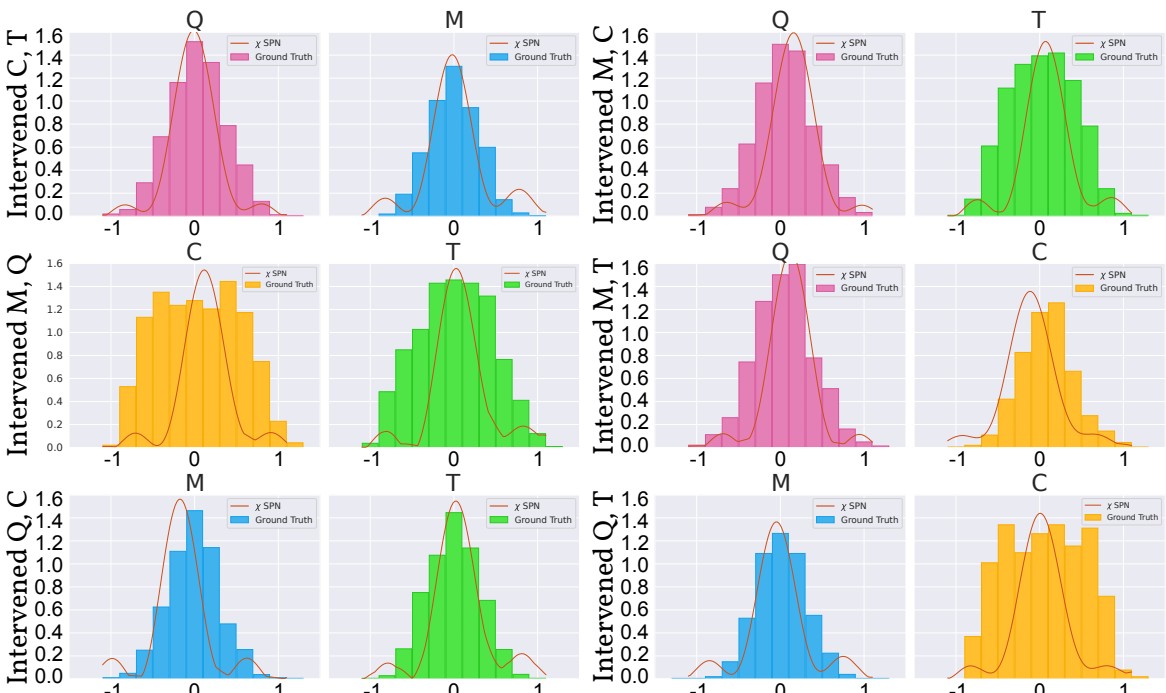

Figure 6: **Generalization to multi-intervention.** Although only trained on single intervention data, $\chi$SPN can generalize to multi intervention estimation. The plots show six combinations of density predictions under two simultaneously applied interventions for the Student data set. As with the single intervention case, modes are generally matched well or offset slightly. Distribution shapes generally match, but show increased errors as distributions flatten out. (Best viewed in color.)

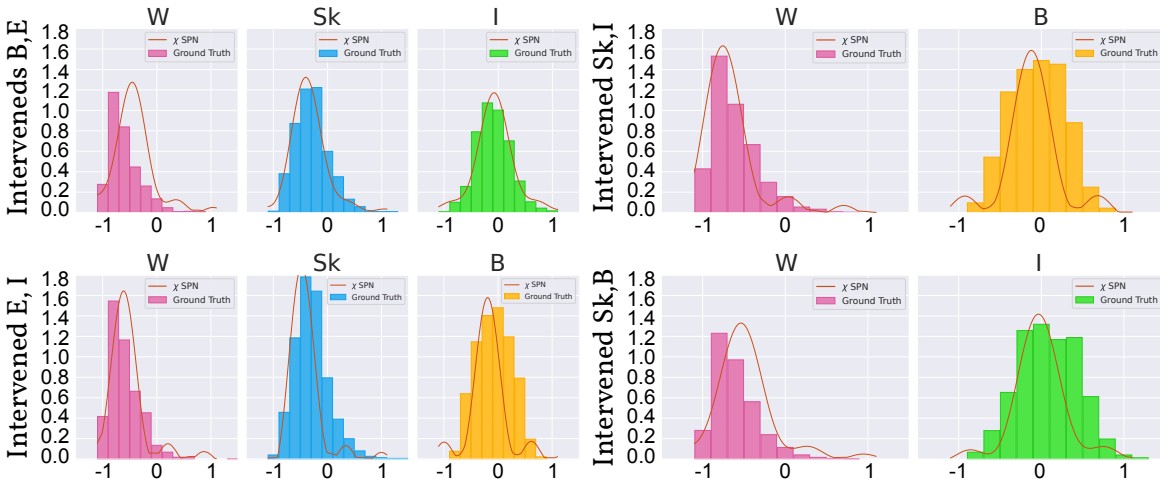

Figure 7: **Multi-Intervention Density Estimation (Hiring Data Set).** The plots show four combinations of density predictions under two simultaneously applied interventions for the Hiring data set. As with the single intervention case, modes are generally matched well with occasional slight offsets. However, in contrast to the Student dataset some intervention combinations (B,E and Sk,B) feature an increased mismatch to ground-truth for '*W*ork Experience'. (Best viewed in color.)

apply [Zhang et al., 2021, Liu et al., 2021].

As far as SPN's are concerned, they can be made robust to out-of-distribution data by modeling uncertainty quantification [Ventola et al., 2023]. This is achieved by a tractable dropout inference (TDI) procedure to estimate uncertainty by deriving an analytical solution to Monte Carlo dropout through variance propagation. Thus, $\chi$SPN can be extended to be more robust towards unseen interventions.

# 5 CONCLUSION

We presented $\chi$SPN, the first causal models that are capable of efficiently inferring causal quantities (i.e., interventional distributions) in presence of mixed data. $\chi$SPN transforms discrete and continuous variables into a shared spectral domain using characteristic functions in the leaves of the interventional SPN. This enables $\chi$SPN to capture the interventional distributions effectively. In addition, we show that $\chi$SPN are able to generalize to multiple interventions while being trained only on a single intervention data thereby showing the generality of our proposed approach.

As most real-world data is mixed by nature, immediate future work includes testing the $\chi$SPN on such real world data sets. Incorporating rich expert domain knowledge, alongside observational data, proves crucial for achieving robust causal inference. Extending our method to integrate such expertise becomes imperative for enhancing the accuracy and reliability of causal models. Also, scaling $\chi$SPN to very large data sets is essential for their adaptation to complex real world scenarios.

## ACKNOWLEDGEMENTS

The TU Darmstadt authors acknowledge the support of the German Science Foundation (DFG) project "Causality, Argumentation, and Machine Learning" (CAML2, KE 1686/3-2) of the SPP 1999 "Robust Argumentation Machines" (RATIO). It benefited from the Hessian Ministry of Higher Education, Research, Science and the Arts (HMWK; projects "The Third Wave of AI" and "The Adaptive Mind"), the Collaboration Lab "AI in Construction" (AICO) of the TU Darmstadt and HOCHTIEF, and the Hessian research priority program LOEWE within the project "WhiteBox". This work was partly funded by the ICT-48 Network of AI Research Excellence Center "TAILOR" (EU Horizon 2020, GA No 952215). This work was supported by the Federal Ministry of Education and Research (BMBF) Competence Center for AI and Labour ("KompAKI", FKZ 02L19C150). The Eindhoven University of Technology authors received support from their Department of Mathematics and Computer Science and the Eindhoven Artificial Intelligence Systems Institute.

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

# A    DESCRIPTION OF SYNTHETIC HETEROGENEOUS CAUSAL DATASETS

Structural equations to the corresponding graphs shown in Fig. 3 used for the experimental evaluation of this paper.

## A.1    CAUSAL HEALTH CLASSIFICATION

$$A = U(0, 100)$$

$$F = \frac{1}{2}A + \mathcal{N}(10, 10)$$

$$H = \frac{1}{100}\left(100 - A^2\right) + \frac{1}{2}F + \mathcal{N}(40, 30)$$

$$M = \frac{1}{2}H + \mathcal{N}(20, 10)$$

// helper variables (used for brevity of notation)

$$D'_1 := \begin{cases} 0.00108A^3 - 0.08862A^2 + 1.337A + \mathcal{N}(25, 10) & \text{if } A \le 4.09837 \\ \mathcal{N}(5, 10), & \text{otherwise} \end{cases}$$

$$D'_2 := 0.0175F + 0.525M + \mathcal{N}(0, 5)$$

$$D'_3 := 0.00013857A^3 - 0.0135A^2 + 0.2025A + 0.2025H + \mathcal{N}(17.1714, 0.2A)$$

// actual diagnose variables

$$D_{i \in \{1..3\}} := \begin{cases} true & \text{if argmax}(\{D'_i\}_{i \in \{1..3\}}) = i \\ false & \text{otherwise} \end{cases}$$

## A.2    HIRING DATASET

$$Sc = U[0, 9]), \text{Discrete}$$

$$W = \frac{1}{2}\text{ChiSquared}(df = 4)$$

$$E = U[0, 6], \text{Discrete}$$

$$Sk = 0.8 * E + 1.2 * W + \text{Pareto}(a = 2.75)$$

$$B = Sc + \mathcal{N}(0, 1.5)$$

$$I = 3 * Sk - \frac{1}{2}B + \mathcal{N}(0, 4)$$

$$D = \mathbb{I}[3 * I + Sk \ge \text{Cutoff}], \text{Binary}$$

## A.3    STUDENT DATASET

$$Sc = U[0, 4], \text{Discrete}$$

$$Q = \mathcal{N}(2.5, 3) - Sc$$

$$M = \mathcal{N}(10, 3)$$

$$C = 0.8 * Q + 0.2 * M + \text{Pareto}(a = 3)$$

$$T = 0.4 * Q + 0.6 * M + \mathcal{N}(0, 1)$$

$$D = \mathbb{I}[T + C \ge \text{RegionalCutoff}] + \mathbb{I}[T + C \ge \text{NationalCutoff}], \text{ 3 categories}$$

## B LEAF TYPES

Here we describe the leaf types that are used in the $\chi$SPN, by following their definitions in Yu et al. [2023].

**Parametric leaf for continuous RVs.** We can assume that the RV at a leaf node follows a parametric continuous distribution $e.g.$ normal distribution. With this, the leaf node is equipped with the CF of normal distribution $\varphi_{L_{Normal}}(t) = \exp\left(it\mu - \frac{1}{2}\sigma^2 t^2\right)$, where parameters $\mu$ and $\sigma^2$ are the mean and variance.

**Categorical leaf.** For discrete RVs, if it is assumed to follow categorical distribution $(P(X = j) = p_j)$, then the CF at the leaf node can be defined as $\varphi_{L_{Categorical}}(t) = \mathbb{E}[\exp(itx)] = \sum_{j=1}^{k} p_j \exp(itj)$.

**$\alpha$-stable leaf.** In the case of financial data or data distributed with heavy tails, the $\alpha$-stable distribution is frequently employed. $\alpha$-stable distributions are more flexible in modelling $e.g.$ data with skewed centered distributions. The characteristic function of an $\alpha$-stable distribution is $\varphi_{L_{\alpha\text{-stable}}}(t) = \exp\left(it\mu - |ct|^\alpha(1 - i\beta\,\mathrm{sgn}(t)\Phi)\right)$, where $\mathrm{sgn}(t)$ takes the sign of $t$ and $\Phi = \begin{cases} \tan(\pi\alpha/2) & \alpha \neq 1 \\ -2/\pi \log|t| & \alpha = 1 \end{cases}$. The parameters in $\alpha$-stable distribution are the stability parameter $\alpha$, the skewness parameter $\beta$, the scale parameter $c$ and the location parameter $\mu$.

## C OBSERVATIONAL DISTRIBUTIONS

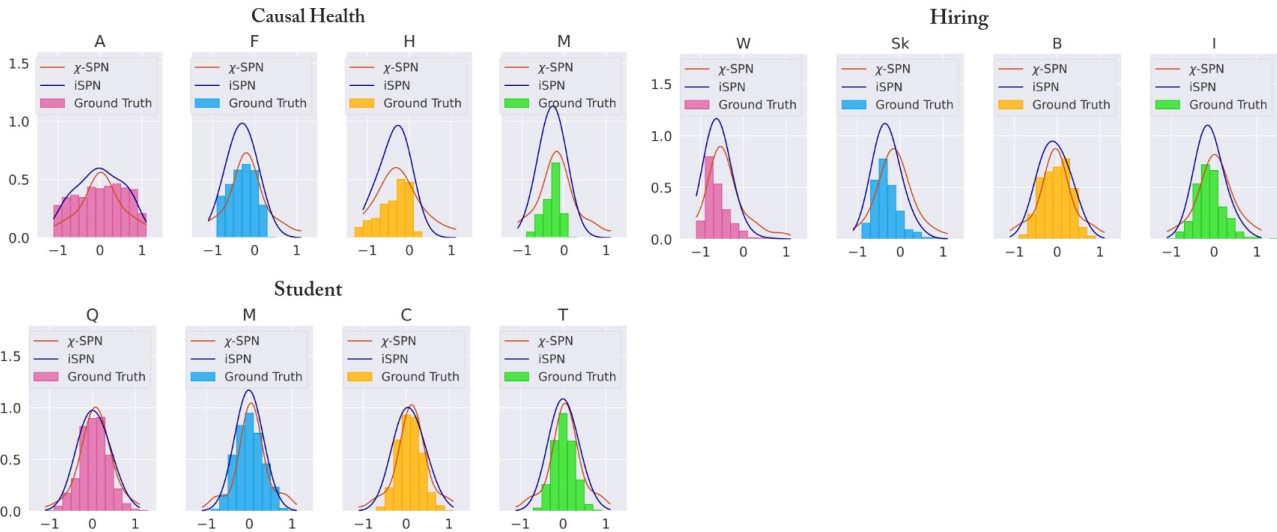

Figure 8: **Approximation of Observational Densities.** Plots feature the approximated densities of continuous variables for all observational data set distributions. Marginalized ground truth distributions (plotted as bar diagrams) and $\chi$SPN approximations (red lines) are shown. Modes of the distributions are generally matched. Within the Hiring dataset (variables W, Sk and I) as well as Causal Health (variables F, H) predicted modes are offset to the ground truth. Discretization artifacts as observed at the boundaries of interventional distributions are strongly reduced. (Best viewed in color.)

# D  χSPN CAPTURES INTERVENTIONAL DISTRIBUTIONS: EXTENDED RESULTS

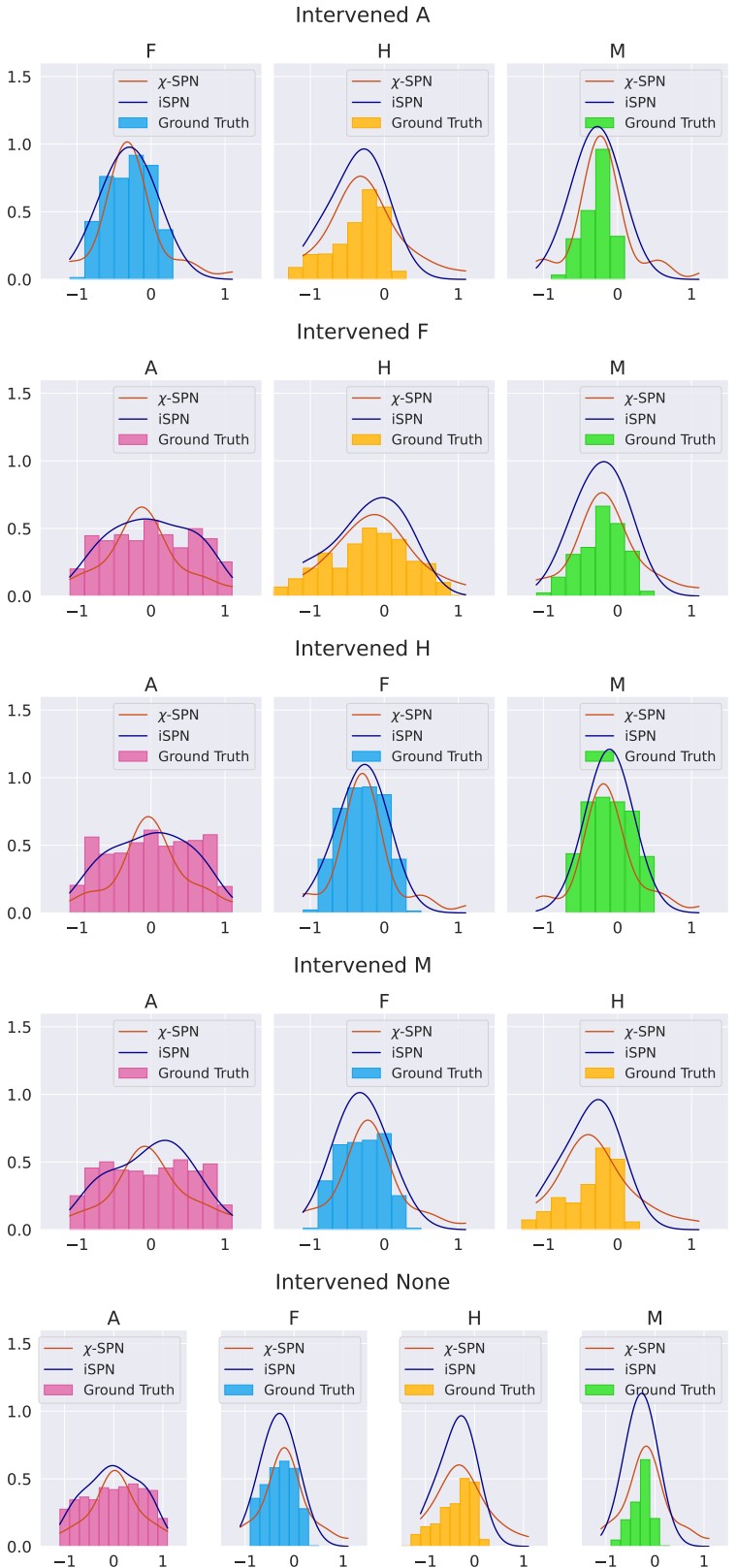

Figure 9: **Approximation of Interventional Densities (Causal Health Data Set).**

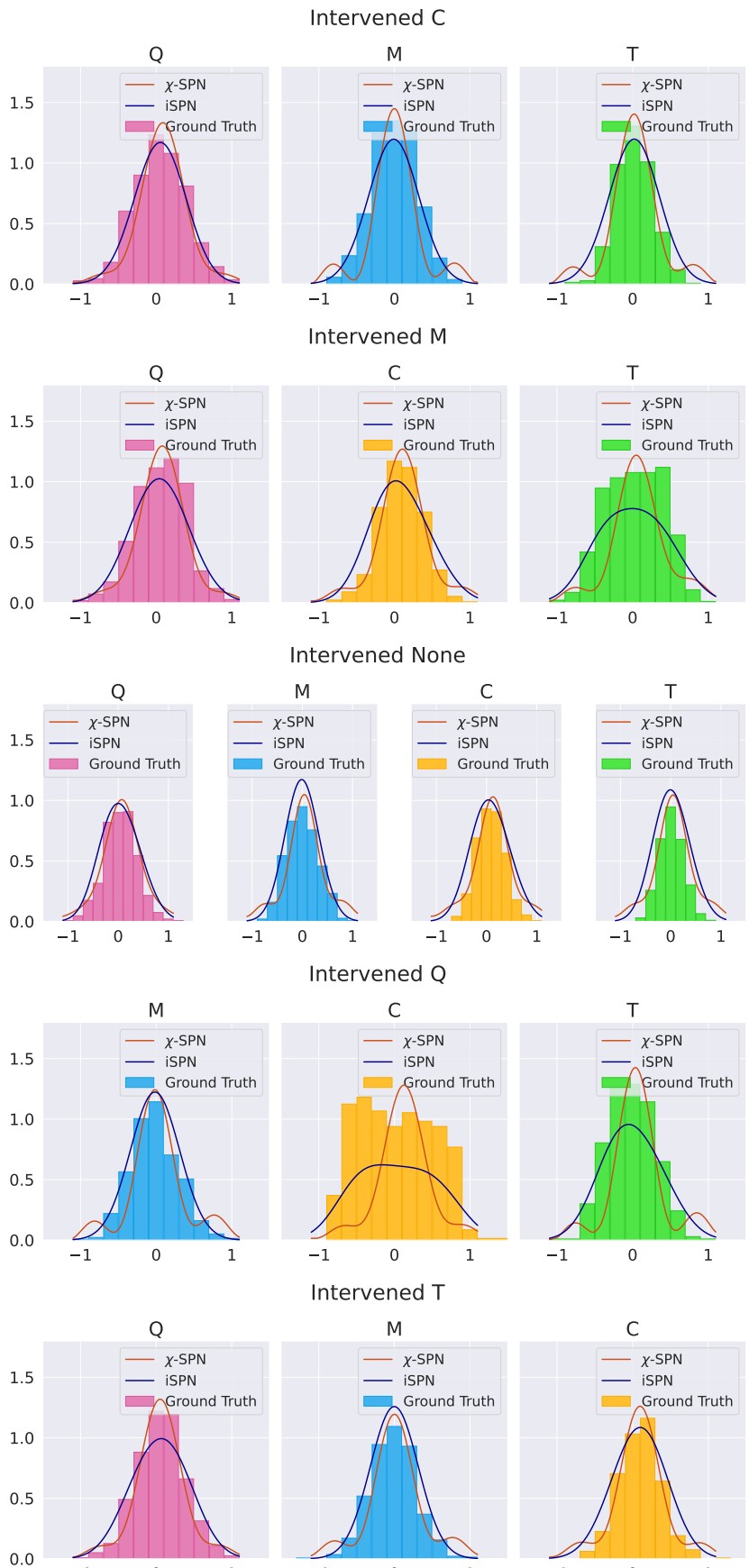

Figure 10: **Approximation of Interventional Densities (Student Data Set).**

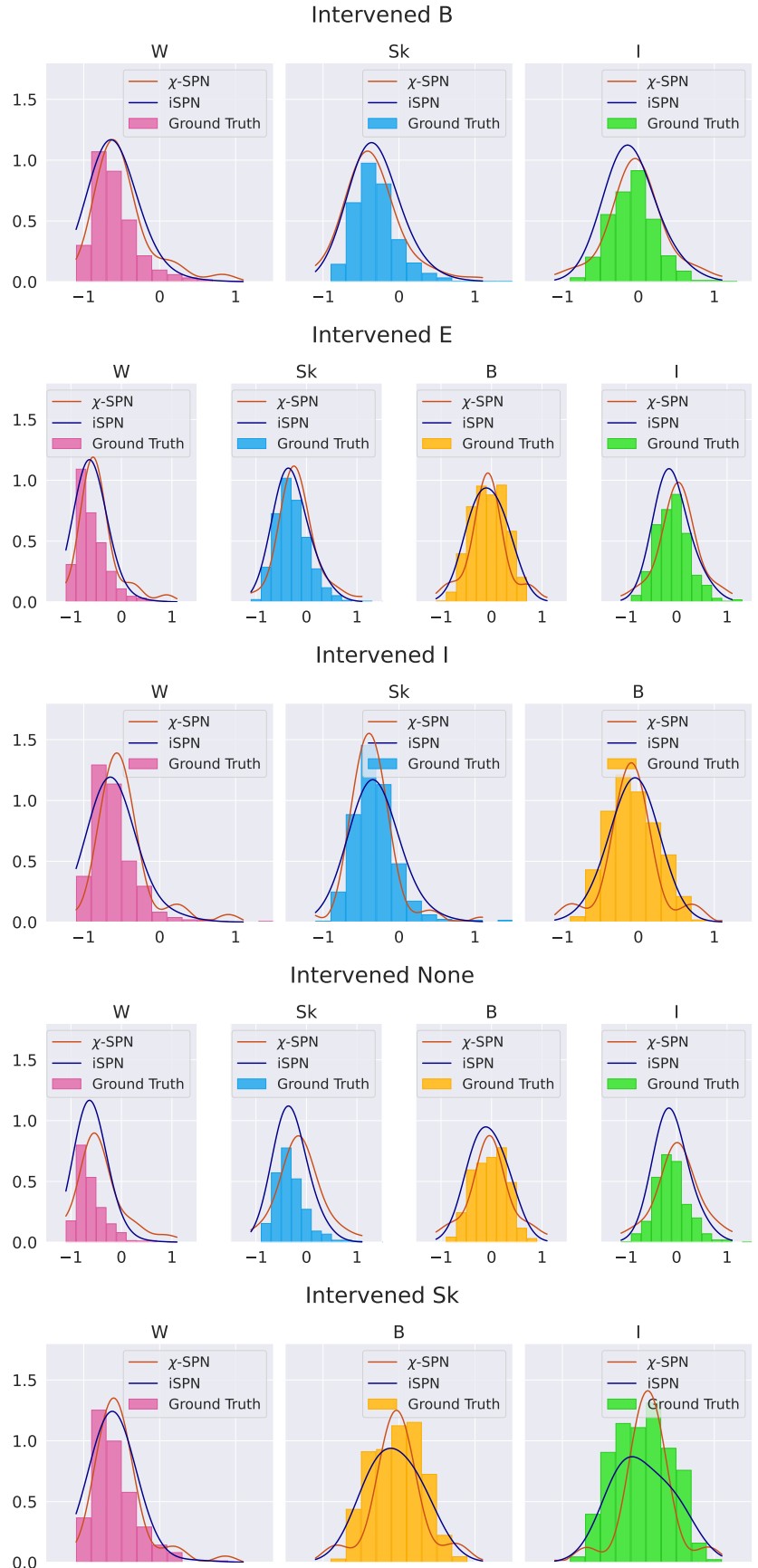

Figure 11: **Approximation of Interventional Densities (Hiring Data Set).**