# OpenReview forum: "$\chi$SPN: Characteristic Interventional Sum-Product Networks for Causal Inference in Hybrid Domains"
_auai.org/UAI/2024/Conference — UAI 2024 poster_

### Official Review · Reviewer_VgNf · 2024-03-15

**Q2-1 Originality-Novelty:** 3
**Q2-2 Correctness-Technical Quality:** 2
**Q2-5 Clarity Of Writing:** 2

**Q10 Ethical Concerns:**

No.

**Q1 Summary And Contributions:**

The manuscipt introduces the Characteristic Interventional Sum-Product Network ($\chi$-SPN) algorithm, which learns a tractable probabilistic model for computing causal quantities. The method is designed for mixed tabular data (i.e., with continuous and categorical covariates). The authors describe some theoretical properties of the method and illustrate its performance on a handful of simulated datasets.

**Q2-3 Extent To Which Claims Are Supported By Evidence:**

2: Fair: the main claims are somewhat supported by evidence (but the experimental evaluation may be weak, or does not match entirely with the claims, important baselines may be missing, proofs contain important ideas but lack rigor, algorithmic details are only discussed superficially, references are imprecise, assumptions are not sufficiently motivated or explicated, etc.).

**Q2-4 Reproducibility:**

1: Poor: key details (e.g. proof sketches, experimental setup) are incomplete/unclear, or key resources (e.g. proofs, code, data) are unavailable.

**Q3 Main Strengths:**

The topic is important and timely. The paper appears to be well researched. $\chi$-SPN builds on existing work in this area and extends it in novel, interesting ways.

**Q4 Main Weakness:**

In several places, the manuscript is unclear and rushed. Key details are omitted, theoretical claims are underdeveloped, and experimental results are hard to interpret.

**Q5 Detailed Comments To The Authors:**

-p. 3: “unintervened $f_i$” is a bit confusing, since this is the structural equation we’re replacing. Perhaps drop the modifier?

-Is $t$ supposed to be a realization of the vector-valued variable $X$?

-In several places, the manuscript claims that deep neural networks are nonparametric. This is not the case.

-$\chi$-SPN appears to require interventional training data as input (in addition to the true underlying graph). From the experiments, however, it appears that the model can (at least in some cases) generalize to unseen regimes. This raises immediate questions of identifiability – under what conditions is such generalization possible? The question matters since a limitation to regimes already in the training data would render the method largely moot. If we already have experimental data for some intervention, then why do we need $\chi$-SPN to perform inference? And if this method can extrapolate to new environments it would be nice to know when/why/how.

-I’m confused about how to compute characteristic functions on discrete data. The definition in Eq. 3 involves integration over $X \in \mathbb{R}^d$. Do we just replace this with a summation for discrete data? What about mixed covariates? This is clearer to me for the empirical case (Eq. 8), where we just run over the values in our data. My impression is that discrete data is in fact being modelled with continuous densities (Lemma 3.2). I find this confusing and somewhat unnatural. How does this method handle gaps in the support of the distribution?

-Does it matter how discrete data are represented (e.g., one-hot encoding vs. using a single column)? I suspect this choice will impact results?

-I was surprised at the mention of MCMC samples following Eq. 11, as this appeared to come out of nowhere. What model is used for this? How do we learn it for generic datasets?

-The experiments fail to benchmark against a single other TPM. This is a remarkable omission.

-I see no proofs in the appendix or code with this submission. This makes it very difficult to evaluate the rigor of the theoretical or empirical results.

**Q9 Complying With Reviewing Instructions:**

Yes

---

> ### Author Rebuttal · Authors · 2024-04-04
>
> We would like to thank the reviewer for their comments. We will respond to the raised questions next.
>
> >  p. 3: “unintervened $f_i$” is a bit confusing
>
> With “unintervened $f_i$” we attempted to emphasize that the original equation in question was replaced. We agree that the wording might be confusing and have adjusted accordingly.
>
> > Is $t$ supposed to be a realization of the vector-valued variable $X$?
>
> $t$: For a vector random variable X, $t \in R^d$ is the argument of the corresponding Characteristic function.
>
> > Incorrect claim that deep neural networks are nonparametric.
>
> Thanks for pointing this out. You are correct, of course. We have corrected that.
>
> > Questions of identifiability – under what conditions is such generalization possible?
>
> This is an important question to be answered in more detail: The major benefit of employing $\chi$SPN is the ability to enable tractable marginal inference. Marginal inference in BNs (and therefore causal BNs / SCM) is #P-hard in general (/NP-hard for approximate inference) (Eiter and Lukasiewicz 2002). Therefore, even when being in possession of a fully specified SCM, it might be beneficial to train a SPN.
>
> The inclusion of interventional data is indeed the effect of your concerns about identifiability. By providing interventions on all variables we enable the NN to correlate the intervention inputs with the corresponding change in SPN weights. As the joint probability of a SCM usually decomposes into a factorized representation, we assume the $\chi$SPN to utilize these independencies. Given modularity due to factorization, errors might occur in cases of context dependent behavioral changes of the structural equation (e.g. interventions pushing values out of distribution; or having unobserved confounders). For these cases no neural model will be able to learn this (all that do --e.g. Neural Causal Models-- assume full sample coverage on the domain in the limit). Thank you for raising these points, again. We will try to highlight this connection about identifiability and generalization at the start of our paper. In general, we are interested in extending future works in this specific direction.
>
> > How to compute characteristic functions on discrete data? How does this method handle gaps in the support of the distribution?
>
> CF for discrete RVs is indeed computed with a summation.
> $\varphi (t) = \sum_{j=1}^k p_j exp(i t j)$ where p_j = P(X=j), j=1…k and k is the number of possible categories.
>
> > Does it matter how discrete data are represented?
>
> It does not matter how discrete data is represented. We employ categorical distribution at leaf nodes for discrete random variables, therefore if the data is one-hot encoded, we can always first convert it to a single column.
>
> > MCMC
>
> The MCMC sampling comes from the expectation in eq(10), which is originally from the integration in eq(9). The MCMC sampling is employed to compute the integration in the defined CFD in eq(9). We assume w(t; \eta) in eq(9) to be a probability density function and in our case a standard Gaussian, therefore we can sample t from a standard Gaussian to approximate the integration, which results in the MCMC sampling in eq (11).
>
> > No baseline is provided
>
> We indeed agree with the reviewer that the results can be a bit hard to judge without a baseline. Please do note that ours is the 1st causal model for mixed data and thus a clear baseline does not exist. We now use interventional SPNs (iSPNs) as a baseline as suggested by reviewer ovGN and the results can be found in the anonymous GitHub link that consists of our code. Specifically
>
> - causal health: https://anonymous.4open.science/r/chi-SPN-6480/experiments/visualizations/ispn_baseline/compare_chc.pdf
> - hiring practices: https://anonymous.4open.science/r/chi-SPN-6480/experiments/visualizations/ispn_baseline/compare_job.pdf
> - student performance: https://anonymous.4open.science/r/chi-SPN-6480/experiments/visualizations/ispn_baseline/compare_student.pdf
>
> As it is evident from the plots, $\chi$SPN can approximate interventional densities better than iSPN across all data sets.
>
>
> > Code and proofs
>
> We provide the full code repository for this paper (see the link on p.2 right before Sec. 2; https://anonymous.4open.science/r/chi-SPN-6480).
>
> We hope we have alleviated your concerns. We will be happy to answer any further questions and look forward to discussing them with you to make our paper better.
>
> **References:**
>
> Eiter and Lukasiewicz. "Complexity results for structure-based causality." Artificial Intelligence 142.1 (2002): 53-89.

---

### Official Review · Reviewer_XaFN · 2024-03-21

**Q2-1 Originality-Novelty:** 2
**Q2-2 Correctness-Technical Quality:** 3
**Q2-5 Clarity Of Writing:** 3

**Q1 Summary And Contributions:**

The paper proposes chi-SPN, a model class used to to express probability distributions that are used to answer causal queries for domains that contain discrete and continuous variables. This is done by combing to recent extensions of standard SPNs [1]. To be specific iSPN used for causal reasoning with discrete variables [2] and characteristic circuits used for probabilsitic reasoning in the discrete continuous domain [3].

The approach is then evaluated on 3 synthetic datasets.


[1] Poon, Hoifung, and Pedro Domingos. "Sum-product networks: A new deep architecture." 2011 IEEE International Conference on Computer Vision Workshops (ICCV Workshops). IEEE, 2011.

[2] Zečević, Matej, et al. "Interventional sum-product networks: Causal inference with tractable probabilistic models." Advances in neural information processing systems 34 (2021): 15019-15031.

[3] Yu, Zhongjie, Martin Trapp, and Kristian Kersting. "Characteristic Circuits." Advances in Neural Information Processing Systems 36 (2024).

**Q2-3 Extent To Which Claims Are Supported By Evidence:**

2: Fair: the main claims are somewhat supported by evidence (but the experimental evaluation may be weak, or does not match entirely with the claims, important baselines may be missing, proofs contain important ideas but lack rigor, algorithmic details are only discussed superficially, references are imprecise, assumptions are not sufficiently motivated or explicated, etc.).

**Q2-4 Reproducibility:**

4: Excellent: key resources (e.g. proofs, code, data) are available and key details (e.g. proof sketches, experimental setup) are comprehensively described for competent researchers to confidently and easily reproduce the main results.

**Q3 Main Strengths:**

It seems that the paper proposed the first method to perform causal reasoning in the discrete-continuous domain in a principled fashion, that at the same time is capable to provide accurate estimate of the probabilities. This is due to the fact (contrary to usual SPNs) that leaves o not form necessarily tractable base distributions.

**Q4 Main Weakness:**

1) It seems that the paper proposes a rather straight-forward combination of two existing methods.
2) The experimental evaluation is rather limited as it is only done on synthetic data sets.
3) Some statements on the paper are factually incorrect.  (see detailed comments)

**Q5 Detailed Comments To The Authors:**

Factually wrong:
- in section 3 (just before Def. 3.1) it is stated that the paper aims at addressing the problem of methods being limited to alpha stable distributions. However, in Section 3.1 leaf nodes are limited to alpha stable distributions. This seems to be a contradiction.
- in the experimental section it is stated that picking a random structure for the SPN "works as well as [...] learning" a structure. This is not true especially when scaling to larger problems. E.g. density estimation on MNIST.

Typo:
bottom page 6: RT-SPN -> RAT-SPN


Question:
Could the authors comment on the difficulty of combing iSPNs or characteristic circuits. To me it seems like a straight-forward task.

**Q9 Complying With Reviewing Instructions:**

Yes

---

> ### Author Rebuttal · Authors · 2024-04-04
>
> We would like to thank the reviewer for their positive evaluation. We respond to the concerns next.
>
> > Paper proposes a rather straight-forward combination of two existing methods.
>
> We can understand the reviewers point. Although it might look like that but, we would like to point that our contribution is not merely a straightforward combination of the two methods. The training of characteristic circuits and iSPNs are very different, iSPN accepts conditional input about interventions whereas CC does not. In order to adapt iSPNs to the spectral domain, we need the model to learn the joint interventional density, and for that we make the root of the $\chi$SPN learn the characteristic function of the interventional distribution. We cannot simply introduce characteristic leaves in an iSPN and later transform it into density for the purpose of learning with log-likelihood.
>
> > The experimental evaluation is rather limited as it is only done on synthetic data sets.
>
> We agree that we use synthetic data sets but please note that no benchmark data for causal inference in a mixed-data setting exists.  Also, most of the causal inference papers make use of synthesized data sets where the data generation process can be controlled. Running $\chi$SPN on real world data is the immediate future next step as we had mentioned in the 1st line of our Conclusion section.
>
> > Factual incorrect statements
>
> - Contradiction in Section 3.1 (just before Def. 3.1): Sorry for the confusion, we do not address the problem of methods being limited to alpha stable distributions. By `moreover, we are restricted to using only those parametric distributions at the leaves that have a closed form density function. This is true only in the case of special α-stable distributions [Nolan, 2013]`, we want to state that the iSPN can only deal with leaf distributions that have a closed-form density, and in fact only special cases of α-stable distributions have closed-form densities. Therefore, with $\chi$SPN, we are able to model leaf distributions that do not have closed-form densities, e.g. (general) α-stable distributions. Hence, there is no contradiction. We have rewritten this part in the paper.
>
> - Random structure works as well as learned structure in larger problems:  It has been shown in the RAT-SPN paper that random structures do work very well even for density estimation in MNIST, Fashion-MNIST etc. Do note that even one of the simplest greedy structure learner, LearnSPN, scales quadratically in the number of the variables. We can modify our statment to "using a random structure is preferable[..]"
>
> We hope we have alleviated your concerns. Wwill be happy to answer any further questions and look forward to discussing them with you to make our paper better.

---

### Official Review · Reviewer_BHKd · 2024-03-22

**Q2-1 Originality-Novelty:** 3
**Q2-2 Correctness-Technical Quality:** 3
**Q2-5 Clarity Of Writing:** 3

**Q10 Ethical Concerns:**

None.

**Q1 Summary And Contributions:**

The paper introduces Characteristic Interventional Sum-Product Networks (χSPN) for causal inference in hybrid domains with mixed discrete and continuous variables. By leveraging characteristic functions in iSPNs, χSPN provides a unified framework and tractable inference, overcoming limitations of previous methods. Notably, χSPN effectively captures interventional distributions for both variable types and generalizes to multiple interventions without retraining, offering significant advancements in causal modeling.

**Q2-3 Extent To Which Claims Are Supported By Evidence:**

2: Fair: the main claims are somewhat supported by evidence (but the experimental evaluation may be weak, or does not match entirely with the claims, important baselines may be missing, proofs contain important ideas but lack rigor, algorithmic details are only discussed superficially, references are imprecise, assumptions are not sufficiently motivated or explicated, etc.).

**Q2-4 Reproducibility:**

3: Good: key resources (e.g. proofs, code, data) are available and key details (e.g. proofs, experimental setup) are sufficiently well-described for competent researchers to confidently reproduce the main results.

**Q3 Main Strengths:**

The main strength of the proposed χ-SPN network is its ability to estimate interventional distributions in the presence of random variables drawn from mixed distributions. The authors demonstrate this capability through extensive experiments. Additionally, the ability to generalize to multiple interventions without any retraining is another notable feature.

**Q4 Main Weakness:**

The proposed approach demonstrates promising experimental results; however, the paper lacks comparisons with baseline methods. Additionally, there is no discussion, analysis, or experimentation on non-asymptotic scenarios, such as how estimation accuracy changes with the number of samples obtained.

**Q5 Detailed Comments To The Authors:**

I have a few questions/suggestions for the authors:

1. Are there any baseline methods that can be used to compare with the proposed approach?

2. How does the number of samples affect estimation accuracy? Conducting experiments to compare accuracy versus the number of samples might help answer this question.

**Q9 Complying With Reviewing Instructions:**

Yes

---

> ### Author Rebuttal · Authors · 2024-04-04
>
> We would like to thank the reviewer for their positive evaluation. We respond to the concerns next.
>
> >No baseline is provided
>
> We indeed agree with the reviewer that the results can be a bit hard to judge without a baseline. Please do note that ours is the 1st causal model for mixed data and thus a clear baseline does not exist. We now use interventional SPNs (iSPNs) as a baseline as suggested by reviewer ovGN and the results can be found in the anonymous GitHub link that consists of our code. Specifically
>
> - causal health: https://anonymous.4open.science/r/chi-SPN-6480/experiments/visualizations/ispn_baseline/compare_chc.pdf
> - hiring practices: https://anonymous.4open.science/r/chi-SPN-6480/experiments/visualizations/ispn_baseline/compare_job.pdf
> - student performance: https://anonymous.4open.science/r/chi-SPN-6480/experiments/visualizations/ispn_baseline/compare_student.pdf
>
> As it is evident from the plots, $\chi$SPN can approximate interventional densities better than iSPN across all data sets.
>
> > How does the number of samples affect estimation accuracy?
>
> We adhere to the training procedure of previous papers (e.g. Zecevic et al.). While we can not run an extensive evaluation due to time constraints, we assume that SPNs --like any other neural model-- will eventually degrade in performance, once training data is no longer able to adequately represent the underlying distribution. We will run this experiment and add to the final version of the paper.
>
> We hope we have alleviated your concerns. We will be happy to answer any further questions and look forward to discussing them with you to make our paper better.
>
> **References:**
>
> Zecevic et al. “Interventional Sum-Product Networks: Causal Inference with Tractable Probabilistic Models.” NeurIPS, 2021.

---

### Official Review · Reviewer_t7PT · 2024-03-23

**Q2-1 Originality-Novelty:** 2
**Q2-2 Correctness-Technical Quality:** 3
**Q2-5 Clarity Of Writing:** 4

**Q1 Summary And Contributions:**

The authors propose Characteristic Interventional Sum-Product Networks ($\chi$SPN), a framework for causal inference in the mixed data setting. This approach stems from the use of sum-product networks, in the mixed-data setting (Yu et al.,2023), and in the interventional causal inference setting (i-SPN) (Zecevic et al., 2021).
Starting from an i-SPN, the characteristic functions are modified, following Yu et al.,2023 in order to take into account mixed-type data. The paper validates this approach using synthetic datasets used in the literature.

**Q2-3 Extent To Which Claims Are Supported By Evidence:**

3: Good: the main claims are supported by convincing evidence (in the form of adequate experimental evaluation, proofs, (pseudo-)code, references, assumptions).

**Q2-4 Reproducibility:**

3: Good: key resources (e.g. proofs, code, data) are available and key details (e.g. proofs, experimental setup) are sufficiently well-described for competent researchers to confidently reproduce the main results.

**Q3 Main Strengths:**

The paper is well written and the contribution is clear. $\chi$SPN is a welcomed extension of the SPNs for interventional causal inference.

**Q4 Main Weakness:**

The experiment section does provide some results, but unfortunately no baseline is provided ; thus making the readers unable to fairly evaluate the contribution.

**Q5 Detailed Comments To The Authors:**

### Questions
- How does your approach react to unseen interventions?
- How does your approach scale w.r.t. the number of variables in the graph?
- How does $\chi$SPN perform in the regressions task (as compared to the classification tasks given in the paper)?

**Q9 Complying With Reviewing Instructions:**

Yes

---

> ### Author Rebuttal · Authors · 2024-04-04
>
> We would like to thank the reviewer for their positive evaluation. We respond to the concerns next.
>
> >  No baseline is provided
>
> We indeed agree with the reviewer that  the results can be a bit hard to judge without a baseline. Please do note that ours is the 1st causal model for mixed data and thus a clear baseline does not exist. We now use interventional SPNs (iSPNs) as a baseline as suggested by reviewer ovGN and the results can be found in the anonymous GitHub link that consists of our code. Specifically
> - causal health: https://anonymous.4open.science/r/chi-SPN-6480/experiments/visualizations/ispn_baseline/compare_chc.pdf
> - hiring practices: https://anonymous.4open.science/r/chi-SPN-6480/experiments/visualizations/ispn_baseline/compare_job.pdf
> - student performance: https://anonymous.4open.science/r/chi-SPN-6480/experiments/visualizations/ispn_baseline/compare_student.pdf
>
> As it is evident from the plots, $\chi$SPN can approximate interventional densities better than iSPN across all data sets.
>
> > Approach reaction to unseen interventions
>
> There are multiple aspects to your question. As the SCM structure is not enforced within the NN, relations have to be learned from a suitable data presentation. Clearly, if a variable is never intervened, the NN can not correlate input of the intervention signal to an appropriate weight vector for the SPN.
> - Assuming that interventions on a variable are trained, but novel values are observed, general assumptions about NNs extrapolating/generalizing to the novel out of distribution values apply. With regard to generalization abilities of SPN we will add a reference to Ventola et al. (2023).
> - Lastly, we have tested generalization abilities of $\chi$SPN to previously unseen combinations of interventions in our experimental evaluation (Figure 5; Sec. 4 Q3), and found that the models generalize fairly well with only slight degradation in performance.
>
> We have added a paragraph on the above discussion to our paper.
>
> > Scaling w.r.t number of variables in graph
>
> Theoretical scaling properties of SPN apply. Specifically, this means $\chi$SPN can perform tractable (marginal) inference in the size of the network (Poon and Domingos, 2011).
>
> > How does SPN perform in the regressions task
>
> This is an interesting question! Generally, SPN are capable of performing regression and classification likewise by estimating the underlying distribution. In that sense SPNs always perform a regression on the distribution. For classification, probabilities for all classes are estimated and the highest mode is picked. Further investigations in the direction of regression with SPN have for example been done by Trapp et al. (2018). We will add a brief clarification on this to our paper.
>
> We hope we have alleviated your concerns. We will be happy to answer any further questions and look forward to discussing them with you to make our paper better.
>
> **References:**
>
> Ventola et al. "Probabilistic circuits that know what they don’t know." Uncertainty in Artificial Intelligence. PMLR, 2023.
>
> Poon and Domingos. "Sum-product networks: A new deep architecture." 2011 IEEE International Conference on Computer Vision Workshops (ICCV Workshops). IEEE, 2011.
>
> Trapp et al. "Learning deep mixtures of gaussian process experts using sum-product networks." arXiv preprint arXiv:1809.04400 (2018).

---

### Official Review · Reviewer_ovGn · 2024-03-25

**Q2-1 Originality-Novelty:** 1
**Q2-2 Correctness-Technical Quality:** 3
**Q2-5 Clarity Of Writing:** 3

**Q1 Summary And Contributions:**

This paper considers the problem of estimating interventional distributions over discrete-continuous hybrid domains. The authors propose a new type of sum-product network, referred to as $\chi$-SPN, heavily influenced by characteristic circuits and interventional SPN (iSPN). Specifically, $\chi$-SPN maps data to the spectral domain similar to characteristic circuits, allowing different mixed discrete and continuous variables to be represented in the same domain; its parameters are learned using the iSPN architecture, where the parameters are given by a neural network conditioned on intervention information. Additionally, the authors show that $\chi$-SPN is a universal function approximator through the use of threshold functions to encode the $\chi$-SPN as a test-arithmetic circuit, which is already known to be a universal function approximator.

**Q2-3 Extent To Which Claims Are Supported By Evidence:**

2: Fair: the main claims are somewhat supported by evidence (but the experimental evaluation may be weak, or does not match entirely with the claims, important baselines may be missing, proofs contain important ideas but lack rigor, algorithmic details are only discussed superficially, references are imprecise, assumptions are not sufficiently motivated or explicated, etc.).

**Q2-4 Reproducibility:**

3: Good: key resources (e.g. proofs, code, data) are available and key details (e.g. proofs, experimental setup) are sufficiently well-described for competent researchers to confidently reproduce the main results.

**Q3 Main Strengths:**

The paper allows for a mapping of variables to the spectral domain, which is new in causal inference to the best of my knowledge. The proposed approach can model a well-defined distribution over mixed variables, even when a closed-form density function does not exist.

The paper is overall well-written and easy to follow.

**Q4 Main Weakness:**

The paper’s contributions are somewhat incremental, as the proposed $\chi$SPN is exactly a characteristic circuit that is parameterized like an iSPN, but it does solve a new problem of estimating interventional distributions over mixed data. I am also concerned about the fact that much of the technical contents describing $\chi$SPN are largely from (Yu et al., 2023) without giving proper credit. For example, Sections 3.1, 3.3, and 3.4 closely follow the sections about characteristic circuit structure, learning, and density computation, respectively, by Yu et al. (2023). The current structure of the paper can thus be misleading in terms of what precisely is the contribution of this work.

The empirical evaluation also lacks comparison to any other baseline. For example, iSPNs can technically handle both discrete and continuous variables. How does $\chi$SPN compare against iSPN? Also, the claim that $\chi$SPN trained with single interventions can generalize to any multi-intervention is quite strong, and is not very convincingly supported by the current set of experiments on synthetic datasets going from single interventions to interventions on two variables only.

**Q5 Detailed Comments To The Authors:**

How are the marginals computed? Section 3.4 shows tractable computation of densities but not marginals.

Does generalization to multi-intervention settings require any additional assumptions in terms of the SCM or data?

Could different interventional distributions by a $\chi$SPN be inconsistent with one another or with the graph structure? For example, in the three-variable graph in Figure 2, will the predictions satisfy P(A | do(B)) = P(A) or P(B | A) = P(B | do(A))?

Minor comments:
- p.6: RT-SPN -> RAT-SPN
- $\chi$ distribution to refer to the distribution given by $\chi$SPN could be confused with Chi distributions.
- Notation: $f$ is quite overloaded, being used to refer to the structural equations, NN parameterizing $\chi$SPN, as well as the density function.
- Is $\omega(t;\eta) >0$ strictly in Eq (9)? This would be important in order to refer to this quantity as a metric.
- Could $w_i,h(t_i)$ in Eq (16) be more explicitly defined? It is a standard result, but this can help readability.

**Q9 Complying With Reviewing Instructions:**

Yes

---

> ### Author Rebuttal · Authors · 2024-04-04
>
> We would like to thank the reviewer for their comments. We will respond to the raised questions next.
>
> > SPN are largely from (Yu et al., 2023) without giving proper credit. Incremental contribution
>
> We apologize for potentially giving the wrong impression. We do give proper credit to Yu et al. (we cite them and explicitly say that we are inspired by them) but will improve. Although we would like to stress that this does not need a major rewrite.
>
> We would also like to point out that using a Characteristic function/circuit in our SPN is beside the point. As the reviewer correctly points out that here we explore new cases for using and generalizing CCs, here causal models over mixed domains. It is priori not clear that this would work well, and solves a much pressing problem, namely, causal inference in mixed domains. Additionally, we would like to point that our contribution is not merely a straightforward combination of the two methods. The training of characterisitic circuits and iSPNs are very different, iSPN accepts conditional input about interventions whereas CC does not. In order to adapt iSPNs to the spectral domain, we need the model to learn the joint interventional density, and for that we make the root of the $\chi$SPN learn the characteristic function of the interventional distribution. We cannot simply introduce characteristic leaves in an iSPN and later transform it into density for the purpose of learning with log-likelihood.
>
> > How are the marginals computed?
>
> For marginal densities, we can perform marginalisation at the input distributions (via inversion at the leaves, as described in Section 3.4) and evaluate sums and products in the network. [On Theoretical Properties of Sum-Product Networks, Section 4 (Peharz '15)]
> For marginal characteristic functions, we refer to Yu ‘23.
>
> >  Does generalization to multi-intervention settings require any additional assumptions?
>
> No. Generalization can be done whenever individual interventions hold information about the multi-intervention case. This might for example be true in the case of additive causal effects of the parents. (Stated differently: It will work, as long as the mechanisms do not start to behave vastly differently for the multi-intervention case; See also our answer Q4 to reviewer VgNf).
>
> > Could different interventional distributions by a $\chi$SPN be inconsistent with one another or with the graph structure?
>
>  If the reviewer is referring to the interventional distributions as generated by the graph mutilation process then no, the data generated is not inconsistent, as it satisfies the properties that can be derived by do-calculus. However, the learned distribution by the $\chi$SPN, can possibly have these inconsistencies since the graph structure is not enforced, rather used to guide the underlying neural network
>
> > how does $\chi$SPN compares to iSPN
>
> Thank you for the comment. We have now added experiments comparing to iSPN and can see that $\chi$SPN can approximate interventional densities better than iSPN across all data sets.
> - causal health: https://anonymous.4open.science/r/chi-SPN-6480/experiments/visualizations/ispn_baseline/compare_chc.pdf
> - student performance: https://anonymous.4open.science/r/chi-SPN-6480/experiments/visualizations/ispn_baseline/compare_student.pdf
> - hiring practices: https://anonymous.4open.science/r/chi-SPN-6480/experiments/visualizations/ispn_baseline/compare_job.pdf
>
> Minor comments:
>
> Thanks for pointing them out. We will correct them in the paper.
>
> - $\chi$ distribution  -> distribution learned by $\chi$SPN
> - make $f$ less overloaded
> - yes $\omega(t;n)$ is strictly $> 0$
> - we will explicitly describe the quantities in eq 16
>
> We hope we have convinced you that even though the text of some sections overlaps with Yu et al. (which is expected since both $\chi$SPN and CCs adapt the characteristic functions), $\chi$SPN is not a simple combination of iSPN with CCs. The sections pointed out by the reviewer are regarding the functions and the leaves, which do remain the same. We hope you can see the utility in our approach. We will be happy to answer any further questions and look forward to discussing them with you to make our paper better.

---

### Meta-Review · Area_Chair_wgGM · 2024-04-21

One expert reviewer (Reviewer ovGn) find the technical contribution somewhat incremental whereas most reviewers agree that the paper could benefit from more baselines and more extensive experimental evaluations. During discussions, Reviewer ovGn also pointed out that "the description of CCs in Sections 3.1, 3.3, and 3.4 have large overlaps with Yu et al. '23 (sometimes almost word-to-word)".

The authors in their rebuttal seem to have provided further experimental results and baselines. And the critical reviewer also acknowledges that the contribution is valid. The authors should rewrite the aforementioned parts of their paper to avoid overlapping writing with Yu et al. as pointed out by Reviewer ovGn.